# Current Status of Management and Outcome for Patients with Ewing Sarcoma

**DOI:** 10.3390/cancers13061202

**Published:** 2021-03-10

**Authors:** Asle Charles Hesla, Andri Papakonstantinou, Panagiotis Tsagkozis

**Affiliations:** Karolinska Institute, 17177 Stockholm, Sweden; andri.papakonstantinou@ki.se (A.P.); panagiotis.tsagkozis@sll.se (P.T.)

**Keywords:** Ewing sarcoma, molecular diagnostics, chemotherapy, surgical treatment, radiotherapy, subsequent primary neoplasms

## Abstract

**Simple Summary:**

Ewing sarcoma is a highly aggressive malignancy affecting primarily children and adolescents. It is the second most common bone sarcoma among children, affecting between 1 and 3 persons per million inhabitants. The tumor typically carries a pathognomonic chromosomal translocation resulting in a fusion transcript (EWSR1-FLI1), which plays an orchestral role in tumor development. While the fusion transcript has been known for decades, targeted treatment has been disappointing. However, new molecular techniques such as next-generation sequencing have significantly increased our understanding of this rare disease. Moreover, new closely related tumors with similar morphology but different characteristics have evolved. Treatment today consists of multiagent chemotherapy, radiation treatment, and surgery, all of which are associated with significant short- and long-term side effects. In this review article, we describe the currently existing diagnostic- and treatment-related challenges as well as the most important ongoing or recently conducted studies.

**Abstract:**

Ewing sarcoma is the second most common bone sarcoma in children after osteosarcoma. It is a very aggressive malignancy for which systemic treatment has greatly improved outcome for patients with localized disease, who now see survival rates of over 70%. However, for the quarter of patients presenting with metastatic disease, survival is still dismal with less than 30% of patients surviving past 5 years. Patients with disease relapse, local or distant, face an even poorer prognosis with an event-free 5-year survival rate of only 10%. Unfortunately, Ewing sarcoma patients have not yet seen the benefit of recent years’ technical achievements such as next-generation sequencing, which have enabled researchers to study biological systems at a level never seen before. In spite of large multinational studies, treatment of Ewing sarcoma relies entirely on chemotherapeutic agents that have been largely unchanged for decades. As many promising modern therapies, including monoclonal antibodies, small molecules, and immunotherapy, have been disappointing to date, there is no clear candidate as to which drug should be investigated in the next large-scale clinical trial. However, the mechanisms driving tumor development in Ewing sarcoma are slowly unfolding. New entities of Ewing-like tumors, with fusion transcripts that are related to the oncogenic EWSR1-FLI1 fusion seen in the majority of Ewing tumors, are being mapped. These tumors, although sharing much of the same morphologic features as classic Ewing sarcoma, behave differently and may require a different treatment. There are also controversies regarding local treatment of Ewing sarcoma. The radiosensitive nature of the disease and the tendency for Ewing sarcoma to arise in the axial skeleton make local treatment very challenging. Surgical treatment and radiotherapy have their pros and cons, which may give rise to different treatment strategies in different centers around the world. This review article discusses some of these controversies and reproduces the highlights from recent publications with regard to diagnostics, systemic treatment, and surgical treatment of Ewing sarcoma.

## 1. Background

Ewing sarcoma (ES) is an aggressive form of sarcoma formerly referred to as Ewing’s family of tumors, which was previously differentiated into classic Ewing’s sarcoma, Askin tumor (ES of the chest wall), and peripheral primitive neuroectodermal tumor (PNET). The terms PNET and Askins tumor are no longer in use, and the group of tumors are now simply classified as ES [1]. Before chemotherapy was introduced in the 1970s, around 90% of ES patients died due to the disease. Today, around 65–75% of patients without detectable metastatic disease at time of diagnosis will survive. Nevertheless, the improvement in survival has plateaued, and the prognosis for the patients with metastatic or for those with primary treatment refractory disease remains dismal [2,3,4].

## 2. Epidemiology

ES is the second most common bone sarcoma after osteosarcoma with an annual incidence of 1–3 persons per million and a peak incidence in the second decade of life [5,6]. There is a slight male predominance, and the incidence is much higher among Caucasians than among African Americans and Asians [7,8,9]. Around 15% of ES arises in the soft tissue, whereas 25% of bony ES occurs in the pelvis and 20% occurs in the femur. If arising in the long bones, the tumor is typically located in the diaphysis [7,10,11,12,13,14]. The lungs are the most common metastatic site (50%), followed by bone (25%) [13,15].

## 3. Clinical Presentation and Diagnosis

Typically, the patients present with pain and swelling, and a history of trauma around the onset of diagnosis is not uncommon. The duration of symptoms prior to the first medical visit is often over 6 months [16]. Patients occasionally have systemic symptoms such as fever or weight loss. An X-ray usually raises a strong suspicion of a primary bone malignancy. Furthermore, an MRI of the tumor including the whole compartment should be performed to determine the extent of bone and soft tissue involvement. Typically, there is a significant soft tissue component of which relation to the vessels and nerves is of central importance when planning for the proper biopsy approach and local treatment.

Definitive diagnosis is usually established through core-needle biopsy, but fine-needle aspiration (in experienced centers) or open biopsy can also be utilized. Morphologically, ES presents as an undifferentiated small round blue-cell malignancy. Mitotic activity is low [17]. The surface antigen MIC2 (CD99) is present in over 95% of tumors, and ES cells usually stain positive for periodic acid–Schiff (PAS) and vimentin [17,18,19]. Molecular diagnosis is of particular importance in the diagnosis of ES since 85% of Ewing sarcomas carry a specific t(11;22) translocation resulting in an EWS-FLI1 fusion transcript [19]. The remaining 15% of cases that lack the EWS-FLI1 fusion transcript usually have a fusion transcript consisting of EWS joined to another member of the ETS family of genes, most often the ERG gene [19,20]. The fusion transcript is routinely detected using fluorescent in situ hybridization (FISH) or reverse transcript PCR (RT-PCR).

Chest-computed tomography is integral part of the staging procedure in order to screen for lung metastases. Detection of bone metastases has traditionally been conducted with bone scintigraphy. However, [^18^F]fluorodeoxyglucose positron emission tomography (FDG-PET) or a fusion PET-CT has replaced bone scintigraphy in most centers as it has proved superior to bone scintigraphy in some studies [21,22,23]. Bilateral bone marrow biopsy and aspirate are also part of routine work-up, but many experts question the additional benefit of bone marrow biopsy and aspirate when the PET scan is negative [21].

## 4. Molecular Diagnostics

Next-generation sequencing (NGS) has become an important diagnostic complement to traditional morphology and immunohistochemistry for mesenchymal tumors and for small round cell tumors in particular [24]. NGS techniques have discovered not only previously unknown ES fusion transcripts not detected by routine FISH or RT-PCR analysis but also new, distinct Ewing-like tumors with different fusion partners, such as the Bcl6 corepressor (BCOR), and capicua transcriptional repressor (CIC)-rearranged sarcomas have been revealed. BCOR rearranged sarcomas, which may have a round-cell or spindle-cell morphologic appearance, harbor fusion transcripts from BCOR gene fusions with CCNB3, MAML3, or ZC3H7B genes [25,26]. ES protocols are generally used in treatment of BCOR rearranged sarcomas, which arise predominantly in older children and young adults and have an explicit male predominance and a prognosis that is as good as, if not better, than ES [25,27,28]. The fusion gene partners in CIS-rearranged sarcomas are most commonly DUX4 and DUX4L and less commonly FOX04. CIS-rearranged sarcomas, which display a round-cell cytomorphology often with a myxoid stroma, arise predominantly in the third or fourth decade, have a strong soft tissue predilection (90%), and less favorable prognosis than BCOR and Ewing sarcomas [25,29]. CIC-rearranged sarcomas constitute a significant portion of round-cell sarcomas lacking EWSR1 rearrangement (60%), whereas BCOR sarcomas only constitute a much smaller portion (4%) [24,30].

Although the majority of ES harbors the oncogenic fusion transcript EWS-FLI1 and EWS-ERG, there is great promiscuity of gene fusion partners in ES [30]. The FET family of genes includes EWSR1, FUS, and TAF15, all of which may form oncogenic gene fusions with one of five members of the ETS genes: FLI1, ERG, ETV1, ETV4, and FEV. Furthermore, the FET family of genes may rarely fuse with non-ETS genes, including the NFATc2, POU5F1 SMARCA5, SP3, or ZSG (also named PATZ1) genes to form what is termed round-cell sarcomas or Ewing like tumors [31,32,33,34].

The combination of morphology and molecular diagnostics has markedly reduced the number of tumors categorized as undifferentiated round-cell sarcomas, which instead has been replaced by specific tumor entities [35]. NGS-based approaches are recommended for classifying small round cell sarcomas where a diagnosis consistent with ES cannot be made following classic immunohistochemistry, FISH, or RT-PCR. Several NGS assays, such as the Archer FusionPlex sarcoma assay, or different NanoString assays are available, but cost is still an issue [36,37,38,39]. Although CD99 is an extremely sensitive marker for ES, CD99 staining is very unspecific as it stains positive in a variety of other tumors [40]. When CD99 staining is weak or when ES cannot be confirmed by FISH or RT-PCR, the panels above will be able to detect ES with more unusual gene fusions, CIC or BCOR-rearranged sarcomas or round-cell sarcomas with non-ETS fusions.

Reliable prognostic markers for relapse in ES are currently lacking, but early data imply that β3-adrenergic receptor (β3-AR) could be a potential candidate. Expression of β3-AR expression, investigated among 20 patients, was higher among patients with metastases and was related to more aggressive course of the disease [41]. Striving for a specific diagnosis and finding prognostic and treatment predictive biomarkers are vital for the choice of proper management and ultimately for survival [35].

Although a genetically homogenous malignancy displaying a very low mutational burden, mutations are seen in the tumor suppressor genes STAG 2, CDKN2A, and TP53 in 17, 12, and 6% of primary tumors, respectively [42,43,44,45]. Mutations or inactivation of STAG2 or TP53 are associated with advanced disease at diagnosis as well as inferior overall survival. Coexisting mutations of STAG2 and TP53 are associated with particularly unfavorable prognosis [42,43,45]. On the other hand, CDKN2A and STAG2 mutations tend to occur mutually exclusively, suggesting a redundancy in alterations in these genes. Moreover, subclonal STAG2 mutated cells have shown clonal expansion in relapsed tumors compared with paired primary tumors, further indicating that STAG2 plays an important role in the metastatic process [43]. Inactivation of these genes can occur either by point mutation, rearrangement, or, perhaps most importantly, by epigenetic factors.

In strong contrast to the genetic homogeneity observed in ES, epigenetic heterogeneity is high. Epigenetic regulation of gene expression may be exerted by DNA methylation, noncoding RNA expression, and particularly by mechanisms of chromatin remodeling. All of these epigenetic actions are geared by the transcription factor EWS-FLI1 [46].

Cell-to-cell studies on cell lines and tumor samples have shown spontaneously existing subclones of cells, constituting a small minority (1.5%) of the total cell population, which display low expression of EWSR1-FLI1. The EWSR1-FLI1^low^ cells are supposedly mesenchymal-like and have a transcriptional signature and phenotype associated with cell migration, invasion, and seeding, properties that are ultimately needed for a tumor to metastasize. On the contrary, the more poorly differentiated EWSR1-FLI1^high^ cells, which by far exceed the EWSR1-FLI1^low^ cells in a tumor, are clearly in a proliferative state, showing robust cell-to-cell adhesion, and lack the migration properties of the EWSR1-FLI1^low^ cells. Over time, ES cells may change from an EWSR1-FLI1^low^ to EWSR1-FLI1^high^ state and vice versa. Intratumoral heterogeneity is thus pronounced in ES, and it is strongly associated with hypoxia and metastasis [47,48].

## 5. Systemic Treatment

The use of chemotherapy has greatly improved the 5-year overall survival rate of patients with localized disease from 10% to around 70% [49,50,51]. Unfortunately, it has had little impact on primary metastatic ES, where 5-year overall survival (OS) is less than 30% [14]. Patients with relapse have an even worse prognosis, with 5-year event-free survival (EFS) rate of only 10% [52,53]. The standard treatment algorithm today is neoadjuvant multi-agent chemotherapy for at least 12 weeks followed by local treatment, which consists of surgery, radiotherapy, or a combination of the two. The duration and type of adjuvant chemotherapy depend on the tumor response to chemotherapy, presence of metastases at diagnosis, and the applied treatment protocol. It is recommended that patients be enrolled in clinical trials when possible.

The Euro-Ewing 2012 trial randomized 640 patients from different countries, seeking to determine whether the European VIDE (vincristine, ifosfamide, doxorubicin, and etoposide) or North American VDC/IE (vincristine, doxorubicin, cyclophosphamide-ifosfamide, and etoposide) regimen is superior. The European protocol consisted of 6 cycles of 3-weekly VIDE, whereas the North American VDC/IE is dose-dense and has longer duration. The North American protocol is based on the Children´s Oncology Group AEWS0031 trial, which demonstrated that induction treatment with alternating cycles of VDC/IE, and IE/VC as consolidation therapy was more effective, and with less toxicity if given every 2 weeks instead of every 3 weeks [54]. The results of the Euro-Ewing 2012 trial have not yet been published. Nevertheless, early results demonstrate improved event-free and overall survival for the VDC/IE group, without increased toxicity [55].

Defining the role of high-dose chemotherapy (HDCT) with stem cell transplant for high-risk patients with localized disease remains an unmet need. This question was addressed in the Euro-Ewing 99 and Ewing-2008 trials. After induction treatment with VIDE, patients with high-risk localized disease were randomized to one course of vincristine, dactinomycin, and ifosfamide (VAI) followed by HDCT with busulfan and melphalan (BuMel) vs. eight courses of VAI. Inclusion criteria for the high-risk arm were poor histologic response after induction treatment (>10% viable cells), large tumor volume (≥200 mL) in patients receiving preoperative radiotherapy or initially resected tumors, and poor radiologic response to induction treatment (<50% reduction of the soft tissue component). However, only 1/3 of the patients fulfilling the eligibility criteria were actually included in the study. Due to slow recruitment, the study was therefore stopped early after recruitment of only 240 patients of the 320 planed [56]. Nonetheless, the HDCT group showed significantly better EFS and OS than patients that only received VAI consolidation treatment [56]. As expected, more toxicity was observed in the HDCT arm. In addition to the low accrual rate, the study had other limitations that should be considered when interpreting the results. Busulfan has known radiosensitizing properties, and is therefore not preferred for patients requiring large radiotherapy volumes or when high doses will be administered to critical organs [56,57]. For this group, which comprises a significant portion of patients in the high-risk group, the alternative is to replace busulfan with treosulfan, a drug in which efficacy is more poorly documented.

Furthermore, the benefit of HDCT after induction chemotherapy with regimens other than VIDE is unclear. Hopefully, the results of the latest Ewing study (Euro-Ewing 2012), which also assigned high-risk patients to Bu/Mel treatment +/− Zoledronic acid, will shed some light into this issue. Although frequently assigned to high-risk patients today, the verdict on the role of HDCT is not yet out [56,58,59,60].

The high frequency of lungs as the first metastatic site and the radiosensitive nature of the disease give rationale to whole-lung radiotherapy (WLRT). However, the recommended whole-lung irradiation dose is limited to 18–20 Gy in 1.8 to 2.0 fractions considering the poor lung tolerance to high-dose radiotherapy [61,62]. Whole-lung irradiation can be considered among patients with lung metastases at diagnosis and in high-risk patients without lung metastases. For the former indication, WLRT has shown some benefits in retrospective non-randomized studies, but the natural limitations of these studies prevent any sound conclusions [63,64,65]. WLRT was still recommended in the Euro-Ewing 2012 trial after completion of consolidation treatment for patients with lung or pleural metastases [66]. When used as a prophylactic treatment for high-risk patients without lung metastases, the term prophylactic lung irradiation (PLI) is used. PLI may have shown a minor benefit in the early trials, where it yielded the same positive benefit as HDCT in preventing lung metastases but with the improvement of systemic adjuvants; PLI is no longer considered for patients with only localized disease [61,65].

Management of patients with metastatic or treatment refractory Ewing is far from established since robust evidence is lacking. Several different polychemotherapy regimens have been used, largely dependent on institutional preferences and based on small studies. The investigators of the international multicenter trial on recurrent and primary refractory Ewing sarcoma (rEECur) have managed a great achievement by bringing together multiple centers in different countries in order to perform a randomized controlled trial to define the standard of care and best backbone chemotherapy regimen. The trial takes advantage of a novel trial design, allowing for new treatment options to be introduced and less effective ones to be rejected. The first and second predefined interim analyses of the trial revealed that gemcitabine/docetaxel (GD) and irinotecan/temozolomide (IT), respectively, were less effective than topotecan/cyclophosphamide or high-dose ifosfamide in terms of tumor response, progression-free survival, and OS [67]. Accrual is therefore now continued only with the latter two treatment options.

## 6. Novel Systemic Treatments

The oncogenic fusion transcripts resulting from fusion of the EWS gene and one of five ETS genes are believed to be essential for tumorigenesis in ES. Targeting the pathognomonic chimeric oncoprotein encoded by the fusion gene has proven to be very difficult. It is therefore with certain optimism that the first-in-class targeted small molecule designed to inhibit the transformation-specific ETS family of oncoproteins, TK216, has shown promising results in a phase I/II study [68]. Partial response or stable disease was observed in 7/11 heavily pre-treated patients receiving the drug as a single agent or in combination with vincristine. For one patient, a regression of all target lung lesions was observed after only two cycles of TK216 alone. Another patient had a partial response with 90% reduction of lung lesions by RECIST 1.1 after two cycles of TK216 and vincristine. The combination of TK216 with vincristine is being further investigated.

Immunotherapy, in any form, has thus far demonstrated discouraging results in bone sarcomas, including ES. Suppression of the immune response by the tumor microenvironment in ES involves different mechanisms, such as expression of myeloid-derived suppressor cells; expansion of immunosuppressive fibrocytes; low expression of human leukocyte antigens (HLAs) A, B, and C; and high expression of HLA-G, which suppresses tumor-specific T-cells [69]. Due to the limited mutational burden and low expression of programmed death ligand-1 (PD-L1), the probability of response to immune checkpoint inhibitors in monotherapy is low [70,71]. The absence of targetable surface antigens limits the applicability of treatment with chimeric antigen receptor T-cell therapy (CAR-T), and the quest for related targets is continuous [72]. Given the characteristics of ES, successful immunotherapy will most likely require a combination with chemotherapy or targeted therapy. The biology of the disease needs to be unraveled to facilitate the choice of proper immunotherapy partners.

Poly (ADP-ribose) polymerase inhibitors (PARP inhibitors) have an effect on BRCA 1 and BRCA 2 mutant malignancies due to inhibition of (PARP1), which is responsible for repairing single-strand breaks. Unexpectedly, ES cell lines have shown a marked sensitivity to PARP inhibitors, despite lacking BRCA mutations. The effect of PARP inhibitors is actually comparable to that of BRCA-deficient cells. Furthermore, the sensitivity to PARP inhibitors is dependent on the EWS-FLI1 translocation being present in the cell lines [73]. Moreover, the EWS-FLI1 transcript (or the EWS-ERG transcript) has been shown to cause accumulation of R-loops (three stranded hybrids composed of a DNA/RNA hybrid and a single-stranded DNA) causing replication stress without inducing homologous recombination. Homologous recombination, which can be mediated through the BRCA1 gene, is important in the DNA repair process. In ES, BRCA1 is trapped by the R-loops, unable to take part in the homologous recombination process. This impairment of BRCA1 gives ES a BRCAness phenotype, making the ES cells dependent on PARP1 for DNA repair in the same way as BRCA-deficient tumors are. Nevertheless, the sensitivity to PARP inhibitors seen in cell lines has unfortunately not been confirmed in patients as the phase II trial, with olaparib failing to show a response in ES patients with refractory disease [74]. Although preclinical data are promising, thus far, clinical benefit has not been equally promising, and a chemotherapy combination is probably more efficient. Therefore, further investigation is required, and PARP-inhibitors are not recommended outside the setting of clinical trials.

Increased replication stress causes genomic instability, which is a tumorigenic hallmark of many malignancies. Ataxia and telangiectasia and Rad3 related (ATR)-protein and checkpoint kinase 1 (Chk1) are important kinases in the DNA damage response machinery that ES relies on in order to survive the increased replication stress [75]. These kinases could constitute potential therapeutic pathways in ES in the future if efficacy is clinically proven [76].

ESW-FLI1 has shown to promote microtubuli stability by enhancing expression of relevant proteins [77]. In addition to vincristine, which forms an important part of Ewing protocols, eribulin has shown preclinical activity in ES and is evaluated in clinical trials (ClinicalTrials.gov identifier NCT03441360).

Temozolamide, a small alkylating agent whose main indication is in treatment of glioblastoma, and irinotecan, a topoisomerase 1 inhibitor, mainly used in treating refractory colorectal cancer, have been used as combination treatment (TEMIRI) for relapsed ES in some institutions [78]. Combination with alkylator-based chemotherapy is currently under investigation in patients with localized disease or metastatic disease (ClinicalTrials.gov NCT01864109). In addition, the role of O6-methylguanine-DNA methyltransferase (MGMT) methylation in TEMIRI activity in patients with refractory ES is examined in an observational study (ClinicalTrials.gov NCT03542097).

Tyrosine kinase inhibitors (TKIs) often target receptors from various tyrosine kinases, but the vascular endothelial growth factor receptor (VEGFR) inhibitors pazopanib, regorafenib, and cabozantinib have gained special interest given the amount of data highlighting the role of angiogenesis in Ewing sarcoma [79]. A retrospective series including two ES patients reported stable disease for 6 and 13 months on pazopanib therapy [80]. Combination of pazopanib with irinotecan and temozolamide proved not to be tolerable in the doses administered in the phase I trial PAZIT [81]. Based on a case report with clinical benefits from pazopanib, the recommendation was made that a Ewing group would be added in the Sarcoma Alliance for Research through Collaboration (SARC) trial 024 (SARC024) [82] investigating the activity of regorafenib in the treatment of liposarcoma, osteosarcoma, ES, and ES-like sarcomas [82]. The primary endpoint for the single-arm, phase II SARC024 trial was PFS at 8 weeks, and 30 patients were enrolled. PFS at 8 weeks was 73% (57–89%), and median PFS was 3.6 months (95% CI 2.8–3.8 months) [83]. Finally, the cabozantinib single-arm phase 2 CABONE trial demonstrated that 26 % (95% CI 13–42) of 39 evaluable patients were progression-free at 6 months [84].

Inhibition of mTOR or CDK4/6 provided encouraging preclinical data, but these have hitherto not been translated to meaningful clinical benefits. Inhibition of mTOR has a clear biological rational given the expression of mTOR in ES [85]. However, rapid development of resistance of mTOR inhibitors warrants investigation of mTOR inhibitors in combination therapies, such as with IGF-R1 inhibitors [86]. Combination therapy seems to be the way forward even for CDK4/6 inhibitors, albeit toxicity, mostly myelotoxicity, remains a hinder. Thus, the use of these agents is preserved within the framework of clinical trials [87]. Selected examples of current ongoing or presented trials investigating novel therapies for metastatic ES are presented in Table 1.

## 7. Local Treatment

It is well known that ES is a radiosensitive tumor. Initially, surgical treatment was confined to expandable bones, but as surgical techniques evolved, surgical treatment indications extended. Reconstruction with modular and expandable endoprostheses, allografts, endoprosthetic–allograft composites, and vascularized autografts are techniques that have been available for long enough to allow for follow-up over 25 years [88]. These procedures have improved functional outcome and enabled limb sparing surgery [88,89,90,91,92,93,94,95,96]. Recycled autografts and segmental bone transport have additionally improved function, facilitating not only limb-sparing but also joint-sparing surgery (Figure 1) [96,97,98]. Computer navigation, intraoperative CTs, and three-dimensional-printed implants are new tools, especially useful in pelvic surgery, which have further improved accuracy in tumor resection and optimized reconstruction [99,100,101,102,103]. Despite advancements in surgical treatment, complications such as post-operative infection, endoprosthetic loosening, and bone healing difficulties are common in this young and active patient group (Figure 2) [104,105].

Radiotherapy, on the other hand, has fewer early complications but serious late side effects, which are becoming increasingly evident as the number of long-term survivors is increasing. The long-term side effects of radiotherapy include growth impairment, insufficiency fractures, and a significantly increased risk for subsequent primary neoplasm (SPN) [6,105,106,107].

Available data favor surgery over definitive radiotherapy in the local treatment of ES [4,106,107,108,109,110,111,112,113,114]. The indications for post-operative radiotherapy are debatable, but it is recommended after intralesional, or perhaps a marginal surgical margin [4,107].

Hence, the current recommendation regarding local treatment of ES is surgical resection with a wide margin. Surgical treatment versus radiotherapy may rarely be a matter of debate when the primary tumor is located in the extremities. However, given the fact that one-third of all ESs are centrally located (in the pelvis and spine), optimal local therapy for the individual patient is commonly debated at multidisciplinary tumor meetings around the world [12]. The discussion usually comes down to whether the tumor can be removed with a clear (wide or marginal) margin, preferably a wide one, without significant morbidity. Since almost all tumors can be resected with a clear margin regardless of location if the associated morbidity and loss of function are ignored, the definition of acceptable surgical morbidity is debatable and individualized. A pelvic tumor in proximity to the acetabulum can in most circumstances be excised and reconstructed with an acceptable risk for complications and a good functional outcome. Most specialists would likely opt for surgery in such a case, but for a pelvic tumor that would require a hindquarter amputation to achieve clear surgical margins, the benefits of surgery are unlikely to outweigh the advantages of radiation treatment. Due to the improved surgical techniques and arsenal of reconstructive options that have developed in recent years, the idea of what is regarded as an operable tumor may have shifted over time.

A randomized study comparing radiotherapy and surgery does not seem feasible given the fact that defined radiotherapy is selected in cases where local excision is not possible or it is accompanied with increased risk for morbidity [4]. Therefore, there is a demand for methodologically sound retrospective studies examining the oncological outcome as well as the late effects with regard to local treatment.

Even though most would advocate surgical treatment over RT, there is still a question to whether definitive RT is inferior to surgery in achieving local control [111,114,115,116,117,118]. In one of the cooperative Ewing sarcoma studies (CESS), a better local control was seen for surgically treated patients compared with patients treated with definitive RT, but no difference was seen in distant relapse, thereby questioning the significance of local control on long-term survival. However, the same study found a significantly lower local or combined (concomitant local and systemic) relapse rate if surgery with wide or radical margins was achieved. This difference was observed regardless of histologic response to chemotherapy. The authors of the study concluded that an adequate surgical margin was beneficial in terms of achieving local control [119]. Another large study based on 982 surgically treated patients from the EURO-Ewing cohort combined with patients in the German Paediatric Oncology and Haematology (GPOH) registry found tumor site and wide surgical margins to be important prognostic factors for local recurrence [120]. On the other hand, a smaller study of 64 surgically treated patients did not find a significant association between surgical margin and local recurrence. Although a trend for inferior local control was seen with surgical margin smaller than 10 cm, the only significant prognostic factor for local control was response to chemotherapy [121]. Nonetheless, the impact of margin on the risk for local recurrence is complex, and the results may reflect the inability to accurately define actual margins. A soft tissue component involving a muscle that shrinks considerably in response to chemotherapy may contain microscopic disease. If this previously involved muscle is not included in the resected specimen and later causes a local recurrence, the local recurrence may falsely be regarded as having occurred despite a previously adequately resected tumor.

A very interesting report was made on the EICESS randomized trial undertaken as a collaborative approach by the GPOH and the Children’s Cancer Leukemia Group (CCLG) in the UK. They found an over 60% OS for the entire group at 5 years, but this concealed a 14% inferior OS among patients in the CCLG cohort. In view of the fact that the baseline characteristics and systemic treatment were the same, the observation raised interest. Further analysis attributed the difference in OS to an inferior local control in the UK cohort where local treatment was clearly less aggressive. The GPOH cohort was more frequently (66 vs. 24%) treated with combined modalities (surgery + RT) than with definitive RT. GPOH patients also received local therapy earlier; 43 and 9% in the GPOH and the CCLG group, respectively, received local treatment within 12 weeks after start of induction chemotherapy [122].

Much attention in European studies like Euro-Ewing has been exclusively on the timing and combination of different chemotherapy regimens. Choice of local treatment is often depended on the physician’s preference and the established local treatment routines. An increased engagement of sarcoma surgeons in these clinical trials is therefore necessary.

Identifying high-risk patients is important in order not only to improve systemic treatment but also to optimize the choice of local treatment strategies. Extent and location of metastases, tumor size and site, patient age, national and institutional practice, and patient preference are all important factors affecting the choice of treatment and subsequently local control and overall survival. For a patient with primary metastatic disease at presentation and a centrally located primary tumor, definitive RT may be the preferred local treatment option. However, surgery may be preferred among patients who respond well to chemotherapy and go into remission of the metastatic disease. Identifying high-risk patients remains a burden given the lack of validated prognostic markers beyond response to preoperative chemotherapy. Tumor size and axial location are generally considered independent risk factors [123,124]. However, large tumors are often axially located and more often treated with definitive RT. Many have therefore questioned the prognostic relevance of tumor size [112,116,125].

Inherent biological characteristics may also identify high-risk patients. Studies have shown that subsets of ES with specific genetic mutations affecting the p53, p16INK4, p14ARF, MDM2, and STAG2 genes are associated with poor response to chemotherapy and thereby poor prognosis [45,125,126,127,128]. Multifactorial risk evaluation and individual case-to-case discussion in multidisciplinary conferences in expert centers are therefore mandated.

The risk of treatment-related subsequent primary neoplasms (SPNs) is an issue that has gained increased interest. Now that more childhood and young adulthood cancer patients survive and are starting to reach an age in which cancer is more common in the general population, we are observing an increasing number of SPNs. Although SPNs are seen after treatment of all childhood malignancies, Hodgkin’s lymphoma and ES survivors belong to the group of patients that have shown the highest risks [129].

Genetic factors as well as treatment-related factors can affect the risk of developing SPNs in patients with bone sarcomas. However, no known inheritable factors are coupled to the increased risk seen in ES [130,131,132]. The chemotherapeutic agents used in treatment of ES, particularly alkylating agents, but also anthracyclines, are known to be carcinogenic, increasing the risk for mainly for hematological malignancies but also for solid tumors, such as breast cancer and osteosarcoma [133,134,135,136,137]. Radiotherapy, which constitutes a well-known risk for developing SPNs, is the only modifiable risk factor [6,106,138].

The literature reports a wide range of risk estimations calculated for SPNs among ES patients. The studies that were uncontrolled and not population-based showed cumulative incidence rates varying from 5% at 10 years to 35% at 10 years [138,139,140,141,142,143]. There are two well-documented large pediatric cohorts, the North American Childhood Cancer Survivor Study (CCSS) and the British Childhood Cancer Survivor Study (BCCSS), both of which have yielded several publications on the risk for SPNs among childhood cancer survivors [144,145]. The cumulative incidences of SPNs among ES survivors in these cohorts were 9 and 10 % at 30 years. In North America, the use of radiotherapy for patients with childhood malignancies has been reduced over the last decades, even so for ES patients [145,146]. The latest CCSS report showed a lower overall cumulative incidence of SPNs among patients treated in the latest (1990–1999) study period, a phenomenon assigned to changes in the use of radiotherapy.

ES survivors are not only at risk for SPNs but also for a wide range of other serious medical conditions [147]. Much work is therefore being put into following these children and young adults long after they have reached the adult world and are considered cured from their primary cancer, surveilling for secondary cancers but also long-term toxicity in terms of cardiac toxicity, renal toxicity, and gonadal toxicity [147]. Therefore, comprehensive long-term follow-up is necessary, and countries can employ different methods depending on the healthcare system available.

## 8. Conclusions

Many cancer patients, particularly children, have benefited from the recent advances in systemic treatment as well as radiation treatment and surgical treatment. However, for patients with ES, the standard of care, consisting primarily of polychemotherapy and local treatment with surgery and/or radiotherapy, has not changed notably in the last decades. Fusion-derived antigens and CD 99 or IGF1R expression could be potential targets for cancer vaccines or CAR-T therapy. Nevertheless, studies investigating the possible role of such treatments, as well as the use of immune checkpoint inhibitors, have, to date, not been promising [148]. Identifying which novel systemic treatment and potential combination is most promising should be the subject of future multinational trials for patients with metastatic or high-risk localized disease.

Low and/or slow accrual in sarcoma trials is a not a negligible problem. This was evident in the Euro-Ewing 2012 trial, which closed in May 2019, and it is also evident in the rEECur study. The latter initiated accrual in 2015 and has only enrolled half of the 525 patients needed. Low accrual in clinical trials is a waste of human and economic resources and most importantly hampers improvement in clinical practice. Initiatives to promote clinical research, as currently being conducted by EU-funded studies and multinational/intercontinental collaborations, such as the Euro-Ewing consortium, are admirable, much needed, and are expected to answer important research questions.

To conclude, improvement to ES management and thereof survival is of paramount importance. Reaching out to potential partners is necessary to achieve prompt results and may solve many problems concerned with funding, regulatory bodies, and the industry. It can also stimulate the inclusion of ES patients in basket trials, which are not restricted to specific cancer types such as the INFORM, where tumor tissue is harvested with the aim of offering individualized treatment for children with recurrent disease regardless of primary type of malignancy [149]. Rare tumors such as ES have a lot to profit from not only clinical collaborations but also interdisciplinary and translational ones.

## Figures and Tables

**Figure 1 cancers-13-01202-f001:**
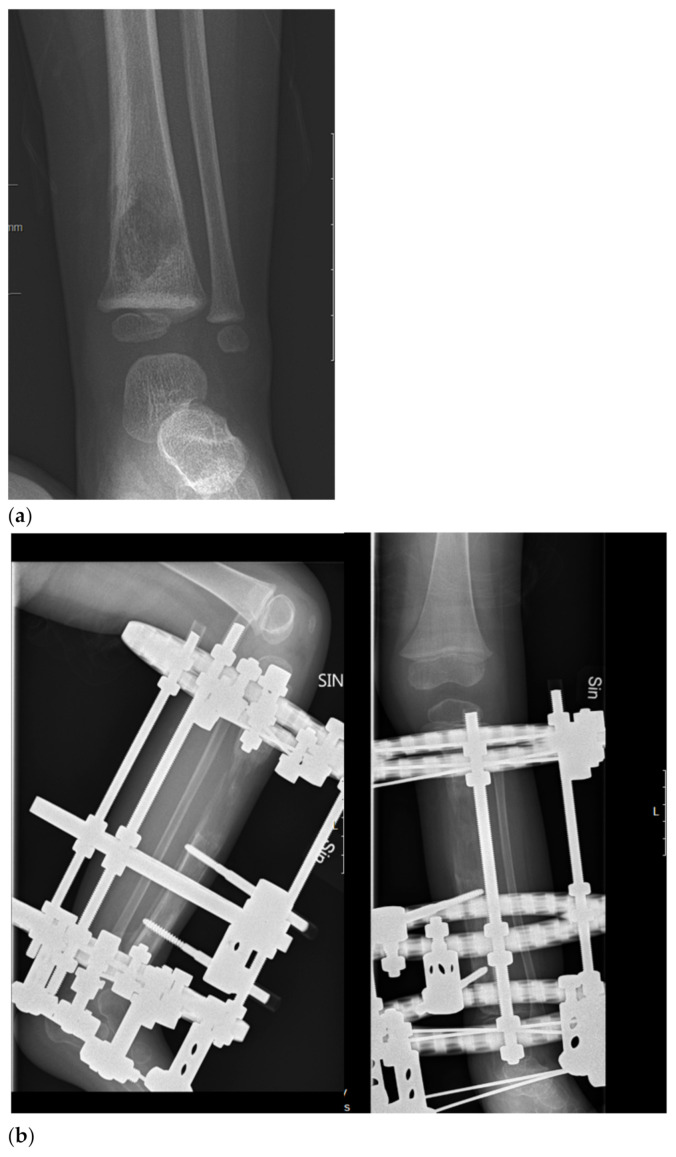
Biological reconstruction using the bone transport technique (**a**) A two-year old girl presented to the emergency department unable to walk on her left leg after a mild fall 3 days earlier. She also had a fever and a c-reactive protein count of 87 mg/liter. X-ray showing a lytic lesion located centrally in the distal tibial metaphysis of the left leg. There is a relatively narrow, but indistinct zone of transition. Periosteal reaction involving the medial aspect of the tibia was observed. Open biopsy showed a dense proliferation of small round blue cells in hematoxylin and eosin. Immunohistochemistry showed membranous positivity for the CD99 marker; staining was also positive for S-100 and periodic acid–Schiff (PAS). Fluorescent in situ hybridization (FISH) demonstrated an EWSR1-FLI1 fusion transcript consistent with the diagnosis of Ewing sarcoma. Staging procedures did not show any metastases. (**b**) Induction chemotherapy with VIDE (vincristine, ifosfamide, doxorubicin, and etoposide) was given according to the Euro-Ewing 2008 protocol. Thereafter, physeal distraction, tumor resection, and segmental bone transport with the use of the Taylor Spatial Frame were carried out. The external fixator was removed after 6 months. (**c**) Five years after removal of the external fixator, the girl is pain free and is able to run and walk without any limitations. Two deformity procedures have been performed after completing oncologic treatment due to a varus deformity secondary to a physeal arrest in the distal tibial physis. She has no evidence of disease.

**Figure 2 cancers-13-01202-f002:**
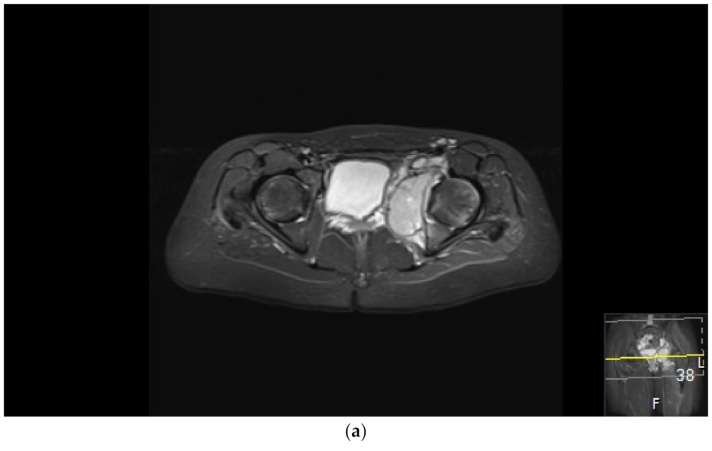
Surgical failure of biological reconstruction after major pelvic surgery, salvaged with endoprosthesis. (**a**,**b**) A previously healthy 15-year old girl with 1-year history of left-sided groin pain. MRI of the pelvis (T1 TIRM coronal (**a**) and axial (**b**) images) showing a bone tumor involving the left superior ramus of the pubic bone and the periacetabular region of the iliac bone. There is a soft tissue component engaging the obturator internus-externus and adductor muscle. Staging procedures did not show any evidence of metastatic disease. Fine-needle aspiration showed a monotonous small round blue-cell tumor most likely representing Ewing sarcoma. FISH analysis showed an EWSR1-Fli1 fusion transcript confirming the Ewing sarcoma diagnosis. (**c**,**d**) After induction chemotherapy with VIDE (vincristine, ifosfamide, doxorubicin, and etoposide), the soft tissue component, as well as the intraosseous extension of the tumor, was significantly reduced, as shown on the coronal (**c**) and axial (**d**) T2 TSE FS MRI images. (**e**) The patient underwent a P2/P3 internal hemipelvectomy, extra-corporeal irradiation with 55 Grey and re-implantation of the autograft. (**f**,**g**) One year after primary surgery, the autograft collapsed (**f**), requiring salvage reconstruction with the Mutars Lumic Cup (**g**). A year later, the patient is functioning well and remains free of disease.

**Table 1 cancers-13-01202-t001:** Selected examples of ongoing, or recently completed, trials investigating novel therapies for metastatic ES.

Drug Class	Example Drug	Number of Patients	Clinical Trial	Results	Current Status	Presented By
EWSR1-FLI1 target agents	TK216 RP2D and expansion	*n* = 15	Phase I/II RP2D and expansion	DCR 47%	Proceeding in further trials	Ludwig et al., ESMO 2020
IGF-1R inhibitors	Ganitumab (added to backbone chemotherapy, metastatic	*n* = 150	Phase III, AEWS1221	No benefit in EFS or OS	Correlative studies	DuBois et al., CTOS 2019
mTOR	Everolimus (Combination with lenvatinib)	*n* = 1	Phase Ib		Phase II ongoing	Dela Cruz, ASCO2020
Microtubuli inhibitors	Eribulin		Phase II NCT03441360		Ongoing	
CDK4/6 inhibitors	Palbociclib, abemaciclib				Are being investigated in early trials	
PARP inhibitors	Talazoparib (combination with irinotecan and temozolamide)	*n* = 22 (both arms)	Phase I	DCR 73%		Federico et al., EJC 2020
Multi-targeted tyrosine kinase inhibitors	Cabozantinib	*n* = 45	Phase II, CABONE	DCR 60%		Italiano et al., ESMO 2018
	Regorafenib	*n* = 23	Phase II, Regobone	DCR 70%		Duffaud et al., ESMO 2020
	Regorafenib		Phase II, SARC024		Ongoing	
	Pazopanib (combination with irinotecan and temozolamide)	*n* = 7	PAZIT	SD 57%	Doses investigated not tolerable	Vo et al., ASCO 2020

CDK4/6: cyclin-dependent kinase 4/6, DCR: disease control rate, EFS: event-free survival, ES: Ewing sarcoma, IGF-1R: insulin-like growth factor 1 receptor, mTOR: mammalian target of rapamycin, OS: overall Survival, PARP: poly (ADP-ribose) polymerase, RP2D: recommended phase II dose.

## Data Availability

No new data were created or analyzed in this study. Data sharing is not applicable to this article.

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
