# Peer review of "Current Status of Management and Outcome for Patients with Ewing Sarcoma"

_cancers, 2021, doi:10.3390/cancers13061202_

Round 1

Reviewer 1 Report

The authors well discuss  the topic,they center the principal argument regarding  Ewing Sarcoma . they make a detailed analysis of both therapies and diagnostic evaluations. I would just have a suggestion. Some  authors recently evaluated the beta 3 adrenergic receptor as a possible additional marker in Ewing's sarcoma. I would suggest that the authors consider this topic in the biomolecular section.

I suggest that the work should be accepted with minor revisions

Author Response

Dear Reviewer,

We thank you for your very useful comment.

Reply to comment: We have now added a paragraph discussing the possible role of the β3-adrenergic receptor as a prognostic marker in line 141 to 146 under the "Molecular diagnostics" section. 

Kind regards

Asle Hesla

Reviewer 2 Report

Overall, this review is well written and focused on the treatment of Ewing sarcoma. I only have a few comments and listed as below:

I encourage the authors to discuss Ewing sarcoma's intratumor heterogeneity because this phenotype may be linked with current treatment resistance. A better understanding of the mechanisms may help develop personalized therapy or precision medicine.

The authors mentioned immunotherapy in the conclusion part. I suggest that the authors discuss how immunotherapy could be a potential treatment method or combination therapy with other agents.

Table 1, why the mTOR inhibitor clinical trial only showed one patient?

Author Response

Dear Reviewer,

We thank you for your valuable comments.

Reply to comment #1: We have now added 2 paragraphs on intratumoral heterogeneity under the section "Molecular diagnostics", line 147 to 172.

Comment # 2: We have now added paragraphs on immunotherapy under the section "Novel systemic treatments" line 260 to 272.

Comment #3:  The reference regards the "phase I/II study of lenvatinib (LEN) plus everolimus (EVE) in recurrent and refractory pediatric solid tumors, including CNS tumors", that was presented as poster at ASCO 2020 Virtual (first auhor Filemon Dela Cruz). Among the 17 patients included, only one had Ewing sarcoma diagnosis and was treated with the escalated lenvatinib dose (11mg/m2) and everolimus 3 mg/m2.

Kind regards

Asle Hesla
